# Evaluation of a New Laboratory Protocol for Newborn Screening for Congenital Adrenal Hyperplasia in New Zealand

**DOI:** 10.3390/ijns8040056

**Published:** 2022-10-21

**Authors:** Mark R. de Hora, Natasha L. Heather, Dianne R. Webster, Benjamin B. Albert, Paul L. Hofman

**Affiliations:** 1Newborn Screening, Specialist Chemical Pathology, LabPlus, Auckland City Hospital, Auckland 1040, New Zealand; 2Liggins Institute, University of Auckland, Auckland 1010, New Zealand; 3Clinical Research Unit, Liggins Institute, University of Auckland, Auckland 1010, New Zealand

**Keywords:** newborn screening, congenital adrenal hyperplasia, CAH, 21-hydroxylase deficiency, LCMSMS, positive predictive value

## Abstract

Between 2005 and 2021, 49 cases of classical congenital adrenal hyperplasia were diagnosed in New Zealand, 39 were detected in newborns and 10 were not detected by screening. For every case of CAH detected by screening, 10 false-positive tests are encountered. Second-tier liquid chromatography-tandem mass spectrometry (LCMSMS) has the potential to improve screening sensitivity and specificity. A new laboratory protocol for newborn screening for CAH was evaluated. Birthweight-adjusted thresholds for first- and second-tier 17-hydroxyprogesterone, second-tier 21-deoxyicortisol and a steroid ratio were applied to 4 years of newborn screening data. The study was enriched with 35 newborn screening specimens from confirmed CAH cases. Newborn screening was conducted on 232,542 babies, and 11 cases of classical CAH were detected between 2018 and 2021. There were 98 false-positive tests (specificity 99.96%, PPV = 10.1%) using the existing protocol. Applying the new protocol, the same 11 cases were detected, and there were 13 false-positive tests (sensitivity > 99.99%, PPV = 45.8%, (X^2^ test *p* < 0.0001). Incorporating the retrospective specimens screening sensitivity for classical CAH was 78% (existing protocol), compared to 87% for the new protocol (X^2^ test *p* = 0.1338). Implementation of LCMSMS as a second-tier test will improve newborn screening for classical CAH in New Zealand.

## 1. Introduction

Congenital adrenal hyperplasia due to 21-hydroxylase deficiency is an autosomal recessive disorder caused by mutations in *CYP21A2* and characterised by the reduced synthesis of aldosterone and cortisol. The loss of negative feedback on the pituitary from cortisol causes increased ACTH secretion and overstimulation of adrenal steroidogenesis that is diverted towards excessive androgen production [1]. There is a spectrum of disease depending on the severity of the mutation in *CYP21A2*. The severest form of CAH results from an almost complete absence in enzyme activity that can lead to life-threatening salt-wasting (SW-CAH) in the first weeks of life and progressive virilisation if not optimally managed over time. The simple virilising form (SV-CAH) has sufficient mineralocorticoid function to maintain electrolyte control under all but the most extreme conditions, but there is progressive virilisation during childhood. The more common, milder non-classic form (NC-CAH) presents later in life with symptoms of androgen excess [2].

Newborn screening (NBS) for CAH has been available in New Zealand (NZ) since 1984, with the primary objective being to prevent the acute clinical symptoms in babies with SW-CAH that usually occur in the first few weeks after birth. The primary screening test involves measurement of 17-hydroxyprogesterone (17OHP) by immunoassay in dried bloodspots collected by heel prick from babies in the first few days after birth. In NZ, the screening test does not distinguish between SW-CAH and SV-CAH as both types are detected by screening and mutation analysis is not part of the screening pathway [3]. Between January 2005 and December 2021, almost one million babies underwent NBS in NZ, with 39 confirmed clinical cases of classic CAH (SW-CAH and SV-CAH) detected in the first weeks after birth. While there are no reported missed cases of SW-CAH (sensitivity 100%) between 2005 and 2021, 10 further children born during the same period, who had not been detected by screening, presented clinically with SV-CAH. The close ties between paediatric endocrinology services and the national NBS programme mean that missed cases of SV-CAH are likely to be reported.

Screening for CAH using bloodspot immunoassay in the early newborn period can result in large numbers of false-positive (FP) tests. 17OHP immunoassays are prone to antibody cross-reactivity that cause falsely elevated measurements, particularly due to the high circulating concentrations of precursor steroids in premature neonates. Additionally, physiological factors such as illness and delayed adrenal enzyme expression in the newborn period can cause elevated blood levels of 17OHP [4]. Before December 2017, a second-tier immunoassay, after solvent extraction to remove interfering polar steroids, was used in NZ to reduce the number of FP tests, but the screening positive predictive value of 1.7% was amongst the lowest of the 23 disorders included in the NZ NBS programme [5].

In December 2017, a second-tier screening test using liquid chromatography—tandem mass spectrometry (LCMSMS) was introduced to improve NBS for CAH in NZ. The measurement of 17OHP, androstenedione (A4), 11-deoxycortisol (11DF), 21-deoxycortisol (21DF) and cortisol (F) is now performed by LCMSMS on the same bloodspot specimen when the initial immunoassay based 17OHP concentration is elevated. LCMSMS is a highly specific analytical technique that is not prone to the antibody cross reactivity interferences that confound steroid immunoassays. LCMSMS implementation using 17OHP alone as a second-tier test resulted in the reduction in FP screening tests and CAH screening positive predictive value (PPV) improving from 1.7% to 11.1% [5].

Further improvements to screening specificity are possible with the addition of second-tier LCMSMS parameters such as using the steroid ratios (17OHP + A4)/F [6], (17OHP)/F [7], 17OHP/11DF [8] or (17OHP + 21DF)/F [9] in conjunction with 17OHP. Laboratories performing second-tier LCMSMS analysis have reported that the FP rate of NBS for CAH was reduced when the ratio (17OHP + A4)/F is applied in conjunction with 17OHP [6,10,11].

The catalytic conversion of 17OHP by mitochondrial P450c11β results in the increased formation of the 11-hydroxylated steroid 21DF, considered a sensitive and specific marker of CAH. When measured by LCMSMS, 21DF can be incorporated as a second-tier marker [12] or be included in a second-tier steroid ratio (17OHP + 21DF)/F calculation [9,13]. The addition of 21DF as a marker could potentially increase the detection of SV-CAH cases in NZ.

The purpose of this study was to evaluate the impact of a revised NBS laboratory protocol for CAH in NZ, using 4 years of retrospective laboratory data and incorporating the second-tier LCMSMS parameters 17OHP, (17OHP + A4)/F and 21DF. To evaluate the potential impact on screening sensitivity for SV-CAH, the study was enriched with residual NBS bloodspot samples from older confirmed CAH cases.

## 2. Materials and Methods

### 2.1. CAH Screening Protocol 1 (Original Protocol)

The primary NBS test for CAH is the measurement of 17OHP using a fluorometric immunoassay (Perkin Elmer, Turku, Finland) on dried blood collected by heel prick between a recommended age of 48 and 72 h after birth. A further sample is collected at 2 weeks from babies born with a birthweight (BW) < 1500 g and a third sample at 4 weeks if BW is <1000 g. Bloodspot specimens with 17OHP immunoassay concentrations above the BW-adjusted cut-off are reflexed to a more specific 17OHP test. A detailed laboratory protocol is shown in Figure 1. On the 1 December 2017, the second-tier 17OHP immunoassay test performed after solvent extraction of the sample was replaced by LCMSMS analysis of 17OHP, A4, 11DF, 21DF and F [5,11].

For babies with a BW ≥ 1500 g, positive screening test results are referred to a paediatrician if second-tier whole blood 17OHP ≥ 63 nmol/L, and a repeat sample is requested by the laboratory if 17OHP is in the range 23–62 nmol/L. For babies with a BW < 1500 g, positive tests are discussed with the paediatric endocrinologist or a senior clinical scientist associated with the NBS programme for appropriate follow-up if 17OHP ≥ 100 nmol/L, while 17OHP results that range between 37 and 99 nmol/L will result in reminder to collect the next scheduled sample. An out-of-range result on the last scheduled screening sample or a result triggering a clinical referral on an earlier sample is considered a positive screen (Figure 1).

### 2.2. CAH Screening Protocol 2 (Revised Protocol)

In January 2022, a new screening protocol was proposed in collaboration with the local paediatric endocrine service (Figure 2). The primary immunoassay screening cut-off remained unchanged for babies with a BW ≥1000 g; however, for extreme low-birthweight babies (BW < 1000 g), the immunoassay screening cut-off was adjusted to ≥72 nmol/L or the 99.8th percentile of all first-tier measurements. The protocol incorporated 3 second-tier LCMSMS markers; 17OHP, (17OHP + A4)/F and 21DF. Cut-offs for second-tier 17OHP were ≥60 nmol/L if BW < 1000 g and remained unchanged in the other BW groups. Birthweight-adjusted screening cut-offs for (17OHP + A4)/F were determined using the 95th percentile calculated in a previous study [14]; these were 2.3 if BW < 1500 g and 1.4 if BW ≥1500 g. A screening cut-off of ≥3 nmol/L was applied to 21DF, above the 2 nmol/L limit of quantification (LOQ).

If the second-tier 17OHP and (17OHP + A4)/F were both above the laboratory threshold or the 21DF was ≥3 nmol/L, the CAH screen result was out of range. All out of range results were considered positive screening tests in specimens from babies with a BW ≥ 1500 g (one sample protocol) or on the last scheduled sample in babies with a BW < 1500 g. The decision point for direct clinical referral was all 3 second-tier parameters above the laboratory cut-off. If 17OHP and (17OHP + A4)/F were above laboratory thresholds but 21DF was below the threshold, results were reviewed by the NBS endocrinologist or senior clinical scientist to determine whether a clinical referral or a repeat specimen was required or, in cases where an additional routine sample was expected, whether the laboratory should wait for the next scheduled sample (Figure 2).

### 2.3. Data Analysis

Four years of NBS data, collected between 1 January 2018 and 31 December 2021 (2018–2021) was included in the study. Laboratory data for all first-tier 17OHP immunoassay and second-tier LCMSMS 17OHP, (17OHP + A4)/F and 21DF during the period of the study was examined. First and second-tier screening cut-offs for each parameter were applied to the data and the FP rate, specificity, and PPV of screening were calculated and compared for protocol 1 (original protocol) and protocol 2 (revised protocol).

To model the effects of protocol 2 on screening sensitivity, the study was enriched with retrospective NBS specimens collected between 2005 and 2017. All NBS samples were stored indefinitely, at room temperature, at an offsite storage facility. Out of 38 retrospective CAH cases, there were 35 NBS specimens with sufficient residual blood for LCMSMS analysis; 25 were collected from babies with classical CAH (SW-CAH or SV-CAH) detected in the newborn period, and 10 were specimens from SV-CAH cases that were clinically detected presenting between the ages of 14 months and 9 years. In total, there were 46 prospective and retrospective specimens included in the study out of a total of 49 cases diagnosed between 2005 and 2021. Of the 10 specimens from cases not detected by screening, 6 did not have any second-tier testing when originally screened, and 4 had testing using a previous second-tier 17OHP method by immunoassay after solvent extraction. For the study, all retrospective samples underwent LCMSMS analysis; however, only 17OHP and 21DF could be evaluated as (17OHP + A4)/F measurements were unreliable due to the instability of cortisol in dried bloodspots specimens in long-term storage at room temperature [15]. Sensitivity for protocol 1 and protocol 2 was modelled using results from all confirmed CAH specimens.

A Pearson’s chi-square test (X^2^) was used to determine the significance of the difference between the sensitivity and specificity of protocol 1 and protocol 2.

## 3. Results

### 3.1. Number of Screening Tests and CAH Cases Detected by Screening (2018–2021)

Between 2018 and 2021, 232,542 babies had NBS for CAH in New Zealand. In total, 236,819 bloodspot specimens were tested, including 4277 that were either tested under the BW < 1500 g sampling protocol or were repeat sample collections. During this time, 11 screen-positive samples were confirmed as cases of CAH, and there were no reports of clinically detected CAH in babies who had received a screen-negative CAH result.

### 3.2. Screening Specificity and Positive Predictive Value (2018–2021)

Applying existing screening cut-offs for the first-tier 17OHP immunoassay for protocol 1, 2403 second-tier tests were performed, of which 1149 were on specimens from babies with a BW < 1500 g and 1254 specimens from babies with a BW ≥ 1500 g. In the BW < 1500 g group, 22% of all specimens were reflexed to second-tier testing compared to just 0.5% of babies with a BW ≥ 1500 g. Applying second-tier cut-offs for 17OHP for protocol 1, there were 230 out of range results; however, 121 of these were encountered in premature babies from whom a subsequent sample was expected, and hence, they were not considered screen-positive tests. There were 109 positive tests, 11 which were true positive (TP) requiring a direct clinical referral and 98 FP requiring a phone request for an additional bloodspot sample. Of the 98 FP tests, 81 occurred in specimens collected from babies in neonatal intensive care units (NICU) and 17 were collected from babies in birthing units or in the community.

Applying cut-offs for protocol 2 over the same period, 497 fewer second-tier tests (*n* = 1906) would have been performed because of the change in the primary immunoassay cut-off for babies with a BW < 1000 g. Applying second-tier cut-offs for 17OHP for protocol 2, there were 26 out-of-range results. Of those, there were 11 TP screens, 10 of which had all three second-tier parameters above laboratory thresholds. One TP had a 21DF below the screening threshold but was referred due to the very high concentration of 17OHP (Table 1).

Of the 15 remaining out-of-range results using protocol 2, 13 were considered FP results. An out-of-range result occurred in an extremely premature baby at day 14 (BW 750 g), and a third sample was expected at 28 days and was thus not considered a positive test. A further specimen from a term baby in the community had an isolated elevation in 21DF (26 nmol/L) with an isolated elevation in 21DF (26 nmol/L), with both 17OHP (8 nmol/L) and (17OHP + A4)/F (0.13) below the screening threshold was considered suspicious for CAH when data was retrospectively evaluated. The result was referred to a local paediatrician; however, the child has been lost to follow-up and was thus not included as a FP screen.

Of the 13 FP screens encountered using protocol 2, 9 had elevations in 17OHP, with ranges 27–49 nmol/L and (17OHP + A4)/F ranging from 1.44–9.22. A further four specimens had isolated increases in 21DF (range 3–5 nmol/L). There were 12 FP results encountered in specimens from babies in NICU due to illness or prematurity, and just 1 was collected from a baby in the community.

During the period of study, the screening specificity for protocol 1 was 99.96%, compared with >99.99% for protocol 2 (1, *N* = 232,542, X^2^ = 53.0324, *p* < 0.0001). The screening PPV was modelled to improve from 10.1% (protocol 1) to 45.8% using protocol 2.

### 3.3. Screening Sensitivity for SV-CAH

Evaluation of the screening data for 2018 to 2021 revealed 11 TP screens when screening cut-offs for protocol 1 or 2 were applied (Table 1).

When protocol 1 was applied to the 35 retrospective samples, there were 25 TP and 10 false-negative (FN) screening tests. Of the 10 FN tests, 6 would not have proceeded to second-tier testing and 4 would have reflexed to second-tier LCMSMS 17OHP measurements below the cut-off of 23 nmol/L. When protocol 2 was applied to the retrospective samples there were 29 TP and 6 FN screening tests. As with protocol 1, six samples would not have proceeded to second-tier testing, but four cases not detected by second-tier testing (17OHP Immunoassay) using protocol 1 had elevated levels of 21DF by LCMSMS and were considered positive tests using protocol 2.

The analysis of the retrospective and prospective data revealed that all specimens were screened using BW ≥ 1500 g laboratory thresholds, although two were very pre-term babies born with a GA of 32 weeks (Table 1 and Table 2). One baby (male), detected by NBS, was diagnosed with SW-CAH, and a second baby (female) was not detected by either protocol.

Incorporating the laboratory data from all 46 CAH cases from 2005 to 2021, 36/46 CAH cases were modelled to be detectable through protocol 1, as compared with 40/46 cases through protocol 2 (Table 3).

Overall, the screening sensitivity of SW-CAH remained at 100% and was unaffected by protocol changes. The screening sensitivity for classical CAH (SW-CAH and SV-CAH) was 78% for protocol 1 and increased to 87% when protocol 2 was applied (1, *N* = 46, X^2^ = 2.24, *p* = 0.1338).

## 4. Discussion

We have developed a new laboratory protocol potentially improving NBS for CAH in New Zealand. The protocol introduces two additional LCMSMS second-tier parameters that reduces FP screening rates whilst improving screening sensitivity. In a previous 2-year study, the PPV of NBS was 11.1% when 17OHP by LCMSMS was used as a second-tier test [5]. In this study, the PPV is lower at 10.1% but improves to 45.8% with protocol changes. Additionally, sensitivity of screening is modelled to improve from 78% to 87%. Of note, this improvement in sensitivity relates only to those with SV-CAH and the sensitivity in SW-CAH remained at 100% for both protocols. Using the new protocol, one FP test result is expected to be encountered for every case of CAH detected, compared with 10 FP results using the original protocol. The new protocol will reduce the number of requests for additional bloodspot samples and associated healthcare costs of follow-up, sample collection, transport, and laboratory processing. Furthermore, we expect almost all FP results to occur in premature or unwell babies within NICUs. This is an important clinical outcome, as it will reduce the anxiety felt by families and community midwives, caused by FP screen results in babies who are at home [16].

Our approach to the introduction of a second-tier LCMSMS test was to adopt a protocol (protocol 1) which incorporated second-tier LCMSMS 17OHP but was otherwise unchanged. This was performed to minimise the risk of compromising screening sensitivity [5,11,14] whilst facilitating the prospective collection of data for informative second-tier markers.

The new protocol (protocol 2) incorporates the steroid ratio (17OHP + A4)/F and 21DF as additional parameters. Several studies had shown that (17OHP + A4)/F offers improvements to screening specificity [6], but there is a risk of reducing sensitivity unless appropriate cut-offs are established [17,18]. We had previously shown that a higher (17OHP + A4)/F cut-off would reduce the number of FP results in premature and low-BW babies and a lower cut-off in babies with a BW ≥1500 g would reduce the risk of FN screening results [14].

Several studies have suggested that 21DF is a sensitive and specific marker for CAH [12,19]. The results of this study demonstrate that while 21DF appears to be a highly accurate marker of CAH, the sensitivity of 21DF was not 100% for either SW-CAH or SV-CAH. Of note, we observed a confirmed case of SW-CAH which did not have a measurable bloodspot 21DF level despite a markedly elevated level of 17OHP and (17OHP + A4)/F ratio. This suggests that, when using the current LCMSMS method, reliance on the detection of 21DF alone could compromise our primary goal to reduce morbidity and mortality in babies with SW-CAH.

While 21DF is a highly specific marker for CAH, elevated levels can also be seen in unaffected babies. We observed four samples which had 21DF levels above the LOQ of our LCMSMS method but 17OHP and (17OHP + A4)/F levels below the screening threshold. Whilst the retrospective data analysis meant that we did not obtain follow-up samples from these cases, the babies were assumed to be unaffected. These findings are consistent with a recent report from the Danish NBS programme, which reported normal follow-up samples in four babies with mildly elevated 21DF levels on an initial sample [20]. Similarly, Held et al. reported that 21DF concentrations in screening samples from unaffected babies ranged from 0.01 to 2.06 ng/mL (0.03 to 6.10 nmol/L) and overlapped with one confirmed CAH case [12]. There are several corticosteroids with the same molecular weight as 21DF, such as 11-DF and corticosterone; however, these are easily separated chromatographically and do not interfere with LCMSMS 21DF results. Further investigations to determine the cause and natural history of mild elevations of 21DF is warranted.

The limitations of 17OHP as a screening marker are well documented [21,22], and with a primary 17OHP immunoassay approach, some cases of SV-CAH will remain undetectable by screening. Minor alterations to immunoassay cut-offs, particularly for babies born at term, may offer further improvements in sensitivity but would also have a negative impact on laboratory workload and costs. The use of 21DF as an alternative primary screening target has been proposed [23]. Should a sensitive and accurate bloodspot 21DF immunoassay test become available, and this may reduce dependency on LCMSMS for accurate CAH screening.

Approximately 75% of second-tier tests were performed on bloodspot specimens collected in NICU during the period of the study. Of those, half were collected from babies with a birthweight of <1500 g. The discontinuation of screening for CAH in premature babies has been recommended due to the poor positive predictive values of screening with an immunoassay approach [24]. Our data analysis revealed one very premature baby boy with SW-CAH was detected by screening while in NICU. Incorporating LCMSMS analysis into screening protocols will increase the accuracy of screening in premature babies in New Zealand and reduce the burdens of follow-up and repeat sample collections. Laboratory costs are higher in screening low-BW babies, but screening is necessary to detect the rare SW-CAH cases that are born prematurely.

Although not part of this study, it is worth mentioning our experience with the costs and laboratory workflow of implementing LCMSMS second-tier testing for bloodspot steroids. All economically developed countries where bloodspot NBS services are available use LCMSMS equipment to screen for inherited metabolic disorders. Modern LCMSMS equipment is now routinely supplied with technology that allows rapid switching between analytical applications that are sufficiently sensitive to measure multiple steroids simultaneously. The LCMSMS equipment used by the NZ programme for bloodspot steroids is shared with an array of clinical applications and NBS has benefitted from the input of experienced LCMSMS scientists. The cost of second-tier testing was estimated at NZD 13.16 per second-tier test or NZD 0.14 (USD 0.09) per baby screened in NZ. Although less easily calculated, implementing LCMSMS testing has led to health cost savings from a reduced number of sample recollections.

Further prospective data analysis over a larger population or longer period will offer a clearer assessment of CAH screening accuracy. Data from the retrospective analysis of steroid levels in stored NBS samples should be interpreted with caution. It has previously been shown that measurements of 11DF and F are not reliable in bloodspots stored longer than 3 months at room temperature [15], and 17OHP has shown to be sufficiently stable for at least 2 years for retrospective studies [25]. We could not identify any similar studies for the long-term stability (>1 year) of 21DF; however, the results were sufficiently informative for the purposes of this study.

In summary, this study demonstrates that the addition of further informative second-tier LCMSMS markers will improve CAH screening accuracy in New Zealand, by increasing both screening specificity and sensitivity. The use of a steroid ratio marker reduces the number of FP tests compared with 17OHP alone and reduces the associated healthcare burden of repeat sample collections and follow-up. Incorporating 21DF is anticipated to detect additional cases of SV-CAH in the newborn period when other second-tier parameters are below screening cut-offs; however, a few additional FP tests will be encountered.

## Figures and Tables

**Figure 1 IJNS-08-00056-f001:**
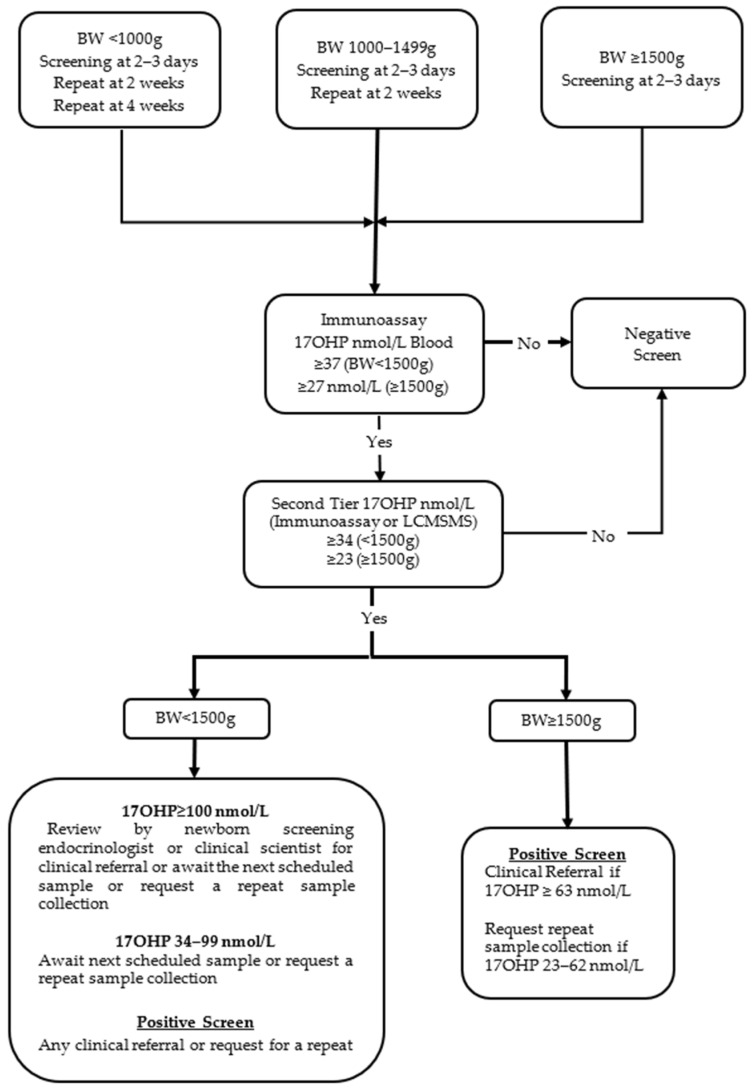
Flowchart of the original laboratory protocol (protocol 1) for NBS for congenital adrenal hyperplasia in New Zealand. Prior to 1 December 2017, second-tier 17OHP measurement was by immunoassay after solvent extraction and was subsequently replaced by LCMSMS.

**Figure 2 IJNS-08-00056-f002:**
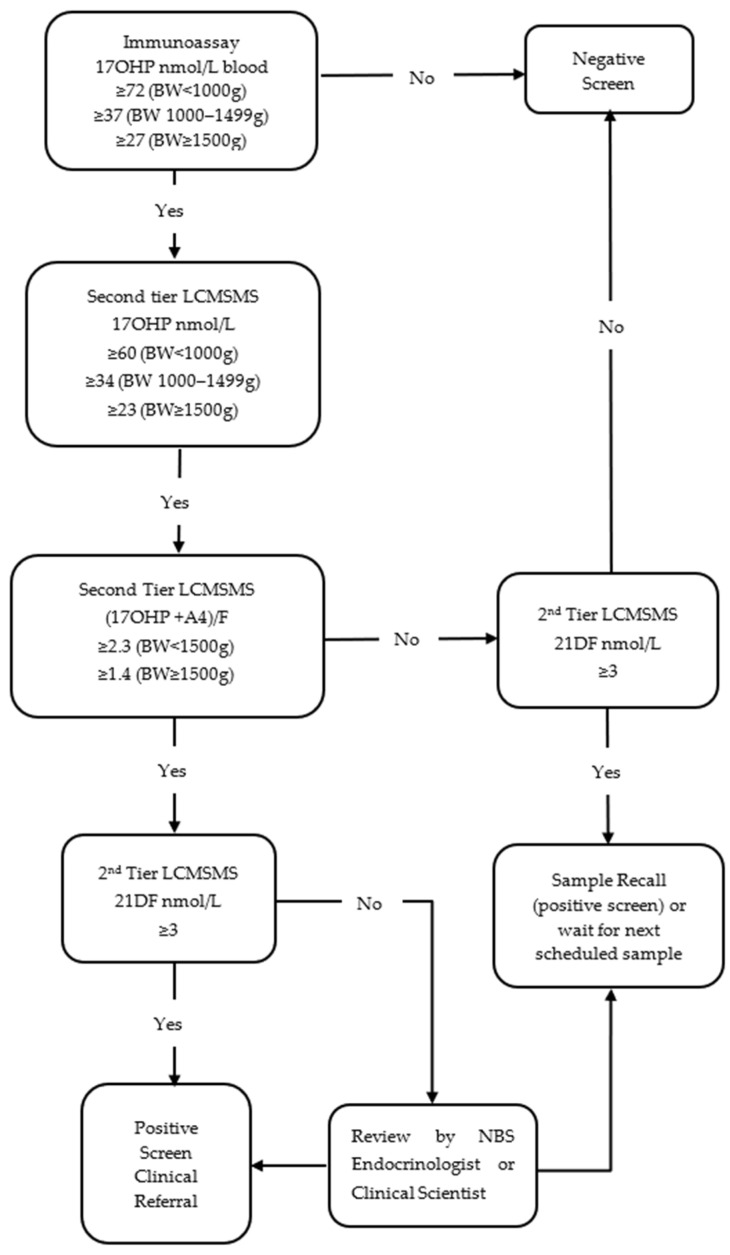
Flowchart of a new laboratory protocol (Protocol 2) for NBS for congenital adrenal hyperplasia in New Zealand.

**Table 1 IJNS-08-00056-t001:** NBS Laboratory Data for 11 confirmed CAH cases 2018–2021. All steroid data is nmol/L blood. All newborn screening results were positive for protocol 1 and protocol.

						LCMSMS Data		
No	Sex	Age (Days)	BW (g)	GA	1st Tier 17OHP	17OHP	(17OHP + A4)/F	21DF
1	M	2	3655	39	715	462	52.6	32
2	F	3	2450	39	599	301	11.9	29
3	F	2	4095	41	358	143	13.6	8
4	F	2	2560	38	338	128	6.8	64
5	F	2	2960	38	627	551	9.8	<2 ^a^
6	F	1	4060	39	667	1422	81.7	88
7	M	2	3810	39	613	383	18.8	39
8	F	2	3890	41	686	833	20.8	73
9	M	2	2902	38	250	112	3.7	22
10	M	2	3975	37	434	138	4.8	25
11	M	2	3670	39	670	761	28.3	64
Laboratory Cut-offs (BW ≥ 1500 g)					**≥27**	**≥23**	**≥1.4**	**≥3**

^a^ The concentration of 21DF was below the screening threshold of 3 nmol/L in case number 5 but would be a positive test under protocol 2, due to the very high concentration of 17OHP and the elevated (17OHP + A4)/F.

**Table 2 IJNS-08-00056-t002:** NBS data from 35 retrospective specimens from confirmed CAH cases in New Zealand in the period 2005–2017. All 17OHP and 21DF concentrations are nmol/L whole blood. For the purposes of the study, a positive screen was second-tier 17OHP or 21DF above laboratory cut-offs.

									LCMSMS Data		
Case No.	Year	Sex	Age	GA	BW (g)	1st Tier 17OHP	2nd Tier 17OHP Immuno	Protocol 1 Screening Result	17OHP	21DF	Protocol 2 Screening Result
12	2017	M	3	37	3010	278	140	Positive	95	5	Positive
13	2014	M	3	41	4330	401	326	Positive	232	6	Positive
14	2014	F	3	41	3750	319	93	Positive	46	3	Positive
15	2013	F	3	38	3610	209	179	Positive	114	2	Positive
16	2013	F	1	36	2580	91	38	Positive	20	19	Positive
17	2013	F	1	41	4200	242	211	Positive	212	209	Positive
18	2012	F	2	40	3650	211	160	Positive	61	14	Positive
19	2012	M	4	40	3165	69	53	Positive	36	10	Positive
20	2012	F	1	40	4100	108	56	Positive	44	11	Positive
21	2011	M	3	39	2970	207	142	Positive	84	10	Positive
22	2010	F	1	34	2410	295	239	Positive	158	34	Positive
23	2010	F	0	38	-	144	111	Positive	76	24	Positive
24	2010	F	1	39	3710	305	131	Positive	73	44	Positive
25	2009	M	2	31	1810	354	305	Positive	194	38	Positive
26	2009	F	5	40	3200	301	201	Positive	268	47	Positive
27	2008	M	4	40	3750	135	66	Positive	42	2	Positive
28	2008	M	3	-	4340	282	63	Positive	39	19	Positive
29	2008	M	2	39	3350	255	186	Positive	97	<2	Positive
30	2008	F	2	41	3605	326	267	Positive	115	66	Positive
31	2007	F	3	-	4360	248	215	Positive	86	38	Positive
32	2007	M	2	41	4330	415	185	Positive	81	21	Positive
33	2006	M	3	40	3620	404	575	Positive	326	9	Positive
34	2006	F	3	42	3230	123	66	Positive	33	48	Positive
35	2005	F	2	39	3340	444	285	Positive	160	12	Positive
36	2005	F	4	-	3760	127	67	Positive	41	16	Positive
**CAH Cases Not Detected by Screening**											
37	2014	M	4	-	4770	30	17	Negative	30	5	Positive
38	2010	F	4	38	2930	72	19	Negative	19	13	Positive
39	2010	M	2	-	3655	34	17	Negative	8	7	Positive
40	2009	F	3	40	3360	30	21	Negative	12	11	Positive
41	2015	F	2	39	3450	5	-	Negative	2	<2	Negative
42	2015	M	3	39	3770	26	-	Negative	12	10	Negative *
43	2012	F	4	39	3640	13	-	Negative	6	<2	Negative
44	2010	F	3	40	3110	6	-	Negative	1	<2	Negative
45	2008	F	5	31	1640	4	-	Negative	2	<2	Negative
46	2006	F	3	40	3530	11		Negative	4	<2	Negative
**Laboratory Screening Cut-offs**						**≥27**	**≥23**		**≥27**	**≥3**	

* Case 42 is a negative screening result using protocol 2 as the specimen would not reflex to second-tier testing.

**Table 3 IJNS-08-00056-t003:** Results for NBS for CAH in New Zealand (2018–2021) (Protocol 1) compared with modelling for Protocol 2. NICU: Neonatal Intensive Care Unit.

	Protocol 1	Protocol 2
**No Second Tier Tests**		
BW ≤ 1500 g NICU	1149	652
BW ≥ 1500 g NICU	765	765
Community or Birthing Units	489	489
**Positive Tests**		
BW ≤ 1500 g NICU	36	3
BW ≥ 1500 g NICU	48	12
Community or Birthing Units	25	9
**False Positive Tests**		
BW ≤ 1500 g NICU	36	3
BW ≥ 1500 g NICU	45	9
Community or Birthing Units	17	1
Total FP Tests	98	13
Screening Specificity	99.96%	>99.99%
Chi Squared Test	X^2^ = 53.0324, *p* < 0.0001	
**Retrospective and Prospective Confirmed CAH NBS Specimens**		
Number of Specimens	46	
Positive Test Results	36	40
Sensitivity	78%	87%
Chi Squared Test	X^2^ = 2.24, *p* = 0.1338	

## Data Availability

The data presented in this study are available on request from the corresponding author, through application to the National Screening Unit, Ministry of Health.

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
