# Peer review of "Evaluation of a New Laboratory Protocol for Newborn Screening for Congenital Adrenal Hyperplasia in New Zealand"

_2409-515X, 2022, doi:10.3390/ijns8040056_

Round 1
Reviewer 1 Report
This paper will be very useful to others working in the field - thank you for sharing.
Some general comments (just to help readers follow as easily as possible - authors are clearly across all the details - some small changes may help readers) are shared for consideration:
1) Abstract - is a bit confusing (line 10 states there were 49 cases of classical CAH, then line 20 states 11 cases of classical CAH were detected). Avoid starting a sentence with number (line 19) - does this number refer to all cases screened between 2005-2021?
2) Introduction - lines 38-39 after "progressive virilisation" perhaps add something like "if not optimally managed over time" (ie virilisation isn't something that has rapid onset in first weeks of life like an adrenal crisis does). Lines 92-93 - details around the additional samples could perhaps be clearer.
3) Methods - details around cut-off levels in 2.1 are hard to read... would a flow chart work? (see comment in results)
4) Results:
- Figure 1 is very useful (unable to read 21DF box because "No" is over top of text) - would it be possible or useful to provide a similar flow chart for the other protocols also? There is a lot of information in the text about the different protocols and cut-off levels and it all gets a bit confusing at times... would inclusion of flow charts for all three (first, interim and revised) protocols be helpful?
- would it be possible to provide more detail in tables 1 and 2, to clarify which #cases are being discussed? For instance, on line 219 where text refers to "6 samples had immunoassay 17OHP levels below the first-tier etc", could numbers of those 6 samples be listed? Also relevant for details given starting line 268 etc.
5) Discussion - really excellent insights.
Overall a really great paper - well done and thank you.
Author Response
Please see attached letter responding to reviewer 1

Reviewer 2 Report
Salt wasting congenital adrenal hyperplasia (SW-CAH) can lead to early neonatal death. This is potentially preventable if the condition is identified through population-based biochemical screening of newborns (NBS), which was first proposed in 1977. The biomarker used then was 17OHP, measured by immunoassay, and it is still used as the first tier approach. For the pas four decades, NBS programs that screen for CAH have been plagued by a very low positive predictive value (PPV). Before 2018, this PPV was 1.7 %, the lowest of the 23 disorders screened for in babies born in New Zealand. The present manuscript describes in detail the changes in the laboratory protocol used between 2005 and 2021 to improve this PPV.
The focus on NBS protocols seems to have distracted the authors from the larger picture. Because I think the case for NBS is strongest for SW-CAH, I wish the paper contained the actual numbers for that subset of patients (lines 158-159 mention 36 CAH cases detected but do not specify how many had SW). Also, false positives are much more common in premature newborns, who may have a lower prevalence of CAH than term infants, an observation that has led the French program to consider discontinuing NBS for CAH in preemies (1, 2). If the New Zealand NBS lab obtains information about gestational age and not just birth weight, this aspect should be mentioned. Along this line, it appears that only 2/46 babies with confirmed CAH had a birth weight < 2.0 kg: how does this compare to the general population of newborns in New Zealand over the period of study?
Minor comments:
-Line 213: Tables should be Table.
-Line 224: delete the word ‘that,.
-Lines 226-227 are unclear to me, could they be rephrased?
1. Coulm B, Coste J, Tardy V, Ecosse E, Roussey M, Morel Y, et al. Efficiency of neonatal screening for congenital adrenal hyperplasia due to 21-hydroxylase deficiency in children born in mainland France between 1996 and 2003. Arch Pediatr Adolesc Med. 2012;166(2):113-20.
2. Huet F, Godefroy A, Cheillan D, Somma C, Roussey M. [Do we need congenital adrenal hyperplasia screening for premature infants?]. Arch Pediatr. 2014;21(2):233-6.
Author Response
please find attached a letter in response to reviewer 2
